# Photocatalytic Oxidation of Propane Using Hydrothermally Prepared Anatase-Brookite-Rutile TiO_2_ Samples. An In Situ DRIFTS Study

**DOI:** 10.3390/nano10071314

**Published:** 2020-07-04

**Authors:** Laura Cano-Casanova, Bastian Mei, Guido Mul, María Ángeles Lillo-Ródenas, María del Carmen Román-Martínez

**Affiliations:** 1MCMA Group, Department of Inorganic Chemistry and Materials Institute (IUMA), Faculty of Sciences, University of Alicante, Ap. 99, E-03080 Alicante, Spain; mlillo@ua.es (M.Á.L.-R.); mcroman@ua.es (M.d.C.R.-M.); 2PhotoCatalytic Synthesis Group, MESA + Institute for Nanotechnology, Faculty of Science and Technology, University of Twente, 7500 AE Enschede, The Netherlands; b.t.mei@utwente.nl (B.M.); g.mul@utwente.nl (G.M.)

**Keywords:** propane photo-oxidation, DRIFTS analysis, anatase-brookite-rutile TiO_2_, propane adsorption, reaction intermediates

## Abstract

Photocatalytic oxidation of propane using hydrothermally synthesized TiO_2_ samples with similar primary crystal size containing different ratios of anatase, brookite and rutile phases has been studied by measuring light-induced propane conversion and in situ DRIFTS (diffuse reflectance Fourier transform infrared spectroscopy). Propane was found to adsorb on the photocatalysts, both in the absence and presence of light. The extent of adsorption depends on the phase composition of synthesized titania powders and, in general, it decreases with increasing rutile and brookite content. Still, the intrinsic activity for photocatalytic decomposition of propane is higher for photocatalysts with lower ability for propane adsorption, suggesting this is not the rate-limiting step. In situ DRIFTS analysis shows that bands related to adsorbed acetone, formate and bicarbonate species appear on the surface of the photocatalysts during illumination. Correlation of propane conversion and infrared (IR) data shows that the presence of formate and bicarbonate species, in excess with respect to acetone, is composition dependent, and results in relatively low activity of the respective TiO_2_. This study highlights the need for precise control of the phase composition to optimize rates in the photocatalytic oxidation of propane and a high rutile content seems to be favorable.

## 1. Introduction

Alkanes, and particularly light alkanes, are pollutants classified as VOCs (volatile organic compounds) that are mainly released to the atmosphere by the use of LPG (liquefied petroleum gases) and as constituents of automobile exhaust emissions [1]. Among them, propane, widely used as a fuel, is one of the most abundant hydrocarbons in air. Environmental protection and the derived legislation imposes actions to develop effective technologies to remove this pollutant from the atmosphere [2], being concentrations in the range of ppm those whose control is more demanding.

Catalytic oxidation is one of the most useful techniques for the abatement of light alkanes. In particular, photocatalytic oxidation has shown to be a suitable solution for air treatment [3,4]. However, limited research has been devoted to the photocatalytic oxidation of propane [5,6,7,8]. In a recent review, Finger et al., discuss the kinetics and mechanism of the photocatalytic oxidation of alkanes and alkenes with TiO_2_ photocatalysts. They indicate that a large electron supply is not sufficient to facilitate a fast propane oxidation, but electrons in appropriate oxygen defect sites seem to be much more important for the catalytic cycle to work [6]. They propose a mechanism in which irradiation is able to produce oxygen vacancies. Localized electrons are supposed to be responsible for propane conversion. Haeger et al., point out that strongly adsorbed hydroperoxide species are formed in oxygen vacancies, and subsequent oxidation of propane appears to be rate determining [9].

In fact, one important aspect of gas-solid photo-oxidation reactions is the formation of strongly bound partial oxidation products, which could poison active sites or even irreversibly react with the catalyst surface. As reported by van der Meulen et al. [7], during photocatalytic propane oxidation, carbonate-carboxylate-formate (R–CO_2_^−^) species are formed on the surface of metal oxides by reaction with hydrocarbons [6,7,10,11]. CO_2_ and H_2_O are formed after reaction of such species with O_2_ [7]. In this mechanism, acetone is a common partial oxidation product and the selectivity towards aldehydes is usually low. This probably means that the alcohol species proposed by Finger et al. [6] can be rapidly oxidized to acetone, which then further reacts under strongly oxidizing conditions (by C_α_–C_β_ bond cleavage and methyl abstraction [7]) to CO_2_ and H_2_O through reaction intermediates like acetate and formate. Based on their spectroscopic findings, van der Meulen et al. [7] propose that adsorbed propane is firstly oxidized to acetone (see Scheme 1) and, then, bridging bidentate species (primarily µ-formate) are formed. They conclude that formation of reaction intermediates is sensitive to the surface structure of TiO_2._ Oxidation of acetone or R–CO_2_^−^ (formate) is rate limiting in conversion of propane over anatase or rutile, respectively [7]. Österlund et al. [10] reported Fourier transform infrared (FTIR) spectroscopy measurements, highlighting the formation of adsorbed bridged acetate species, among others. Szanyi et al. [11] studied the photo-oxidation of acetone on a commercial oxidized TiO_2_ (P25) by in situ FTIR analysis. They found that acetone is completely converted to acetate and formate, and ultimately to CO_2_ and water, and they proposed that bicarbonate/carbonate species can be formed by the interaction of the generated CO_2_ and H_2_O with the solid surface.

Such species are shown in Scheme 1, which has been redrawn combining the schematic pictures established by van der Meulen et al. [7], Österlund et al. [10] and information provided by Szanyi et al. [11].

Previously, we developed a hydrothermal synthesis method to facilitate preparation of TiO_2_ photocatalysts containing mixed crystalline phases (anatase, brookite and rutile) [12]. These mixed-phase photocatalysts showed improved performance for the removal of propene. Considering the importance of removing light alkanes, the present study is devoted to analyse structure–activity relationships for low-frequently explored propane photoxidation using mixed-phases TiO_2_ photocatalysts. A thorough diffuse reflectance Fourier transform infrared spectroscopy (DRIFTS) study aimed to identify surface intermediates involved in the process enables correlation of their nature and abundance with photocatalytic activity. Such a study is of interest because identification of the rate-determining step/s is essential to devise new methods and materials with improved photocatalytic efficiency. As concluded by Busca el al. [13], in situ studies under reaction conditions are certainly very useful to monitor the state of the working catalyst and, in particular, to monitor the presence of strongly adsorbed species during reaction. 

Thus, the novelty of this work, compared with previously reported studies focused on the photoxidation of propane [7,14], relies mainly in the use of a series of photocatalysts with significant structural complexity, due to the coexistence of three titania crystalline phases, particularly the presence of metastable brookite phase, with the subsequent variability of properties linked to structural differences, such as surface area and band gap. This is also the first time that, in addition to an in situ DRIFTS study of the photocatalysts under working conditions, an analysis of the adsorption behavior has been performed.

## 2. Materials and Methods

### 2.1. Materials

Titanium (IV) tetraisopropoxide (C_12_H_28_O_4_Ti, 97%) was purchased from Sigma-Aldrich. Absolute ethanol (C_2_H_6_O, 99.5%) and hydrochloric acid (HCl solution, 37%) were supplied by Panreac. All reactants have been used without further purification. The commercial TiO_2_ P25 from Degussa has been used as a reference photocatalyst.

### 2.2. Preparation of TiO_2_ Materials

TiO_2_ was prepared by a hydrothermal method [12], in which 4 mL titanium (IV) tetraisopropoxide (TTIP) and 20 mL ethanol were stirred at room temperature for 1 h, and then a mixture of HCl (4 mL, 0.5, 0.8, 1, 3, 5, 7 or 12 M) and ethanol (10 mL) was added dropwise. After 1 h of continuous stirring, the solution was transferred to a 50 mL Teflon-lined stainless-steel autoclave and kept at 180 °C for 12 h. After cooling down to room temperature and filtration, the solid was dried (100 °C, 12 h) and then heat treated in air (5 °C/min, 350 °C, 2 h). 

The synthesized materials are named TiO_2_–XM, where X refers to the molar concentration of the HCl solution used. Note that the acid concentration in the synthesis pot is about 10 times lower (i.e., if 4 mL of 12 M HCl are used, and considering that the total volume is 38 mL, the HCl concentration in the synthesis medium is 1.27 M).

### 2.3. Characterization

Surface area and porosity of the prepared samples were determined by N_2_ adsorption at −196 °C (Autosorb-6B, Quantachrome) after outgassing at 250 °C for 4 h. The Brunauer-Emmett-Teller (BET) equation was applied to the adsorption data to calculate the apparent surface area (S_BET_) and the total pore volume (V_T_) was determined from the volume of N_2_ adsorbed at P/P_0_ = 0.99 (P and P_0_ are, respectively, the actual and the saturation pressure of the adsorbate at the temperature of the experiment.) [15].

Crystallinity and phase composition of TiO_2_ samples were determined by X-ray diffraction (XRD), following the analysis method described in previous work [10]. XRD patterns were recorded for pure TiO_2_ samples and TiO_2_/CaF_2_ mixtures (50%, *w*/*w*), (CaF_2_ used as a standard), with Cu Kα radiation (0.1540 nm), at a scanning rate of 2°/min, in the 2θ range 6-80° (Miniflex II Rigaku (30 kV/15 mA). The average crystallite size was calculated by the Scherrer equation (Equation (1)) [16]:(1)$$B=\frac{K\lambda }{\beta cos\theta }$$
where *B* is the average crystallite size (nm); λ is the wavelength of the radiation used, *K* = 0.93 is the Scherrer constant [16], β is the full width at half maximum intensity (FWHM) and θ is the angle associated to the main peak of the studied phase (2θ values of 25.3, 27.5 and 30.8° for anatase, rutile and brookite, respectively).

The optical absorption properties were studied by diffuse reflectance ultraviolet–visible (UV-vis) spectroscopy (Jasco V-670, with an integrating sphere accessory and powder sample holder). BaSO_4_ was used as reference material and the reflectance signal was calibrated with a Spectralon standard (Labsphere SRS-99-010, 99% reflectance, North Sutton, NH, USA). The absorption edge wavelength was estimated as the intercept of the highest slope tangent line of the spectrum with the *x* axis (absorbance method). Then, the band gap was calculated [17] as:(2)$$Eg=\frac{1239.8}{\lambda }$$
where *Eg* is the band gap energy (eV) and λ is the edge wavelength (nm).

### 2.4. Photocatalytic Oxidation of Propane in Batch Reactor

Photocatalytic oxidation of propane was evaluated in a 3 mL batch reactor (see reactor flowsheet in Appendix A) equipped with a quartz window and a 365 nm light-emitting diode (LED) lamp (APGC1-365-E, 135 mW) [18,19]. A gas distribution system allowed for preparation of customized gas mixtures. The outlet stream was analyzed by Gas Chromatography, GC (Agilent 7820, with Varian CP7584 column and a Methanizer- Flame Ionization Detector (FID) combination detector). Photocatalyst films on glass supports (26 mm × 26 mm) were prepared by drop-casting using 750 µL of a photocatalyst suspension (10 mg photocatalyst in 5 mL distilled water) to form a homogeneous coating. The samples were left in a vacuumed desiccator until dryness. 

Prior to the measurements, the batch reactor was flushed (21 min) with 30 mL/min of the reactant gas mixture (4890 ppmv propane, 19.5 vol.% O_2_ and 80 vol.% N_2_). Then, the reactor was closed and, after 10 min, the gas present in the reactor was swept by He flow (10 s) for GC analysis. After a further purge (21 min) with the reactant gas mixture, the reactor was closed, the sample was illuminated and, afterwards, the gas composition was analyzed. This sequence (21 min gas flow, illumination, gas analysis) was repeated for different illumination times (10, 20, 30 and 60 min) (Appendix A, shows these steps schematically).

### 2.5. In Situ Diffuse Reflectance Fourier Transform Infrared Spectroscopy (DRIFTS) Study of the Photocatalytic Oxidation of Propane

In situ DRIFTS measurements were performed using a Bruker Vertex 70 spectrometer equipped with a liquid N_2_ cooled MCT (Mercury-Cadmium-Telluride) detector and a Harrick Praying Mantis diffuse reflectance accessory. A quartz window was used to illuminate the catalyst with a 375 nm LED (Appendix A; maximum light intensity at the catalyst surface of 10 mW/cm^2^), while two ZnSe windows provided an optical path for infrared (IR) analysis.

In each experiment, 30 mg of photocatalyst were exposed to a 2 vol.% propane in air stream (20 mL/min) for 30 min. The cell was closed and, after 10 min, a reference spectrum was recorded. Afterwards, illumination was switched on, one DRIFTS spectrum was immediately recorded, and further spectra were recorded every 5 min for a period of 60 min.

## 3. Results

### 3.1. Photocatalysts’ Properties

Table 1 summarizes the physical properties of the prepared TiO_2_ samples. Commercial P25 is included for reference.

All TiO_2_–XM samples have larger surface area and pore volume than P25 [12]. In general, the surface area concomitantly decreases with an increase in average anatase crystallite size. This higher size is the consequence of the use of higher HCl concentrations for the preparation of the photocatalysts [12]. A high crystallinity is obtained independent of the synthesis conditions (~80%) but, importantly, the phase composition strongly depends on acidity of the precursor solution. Only for the lowest (TiO_2_–0.5M) and highest (TiO_2_–12M) HCl concentrations, dual crystalline phase photocatalysts are obtained, whereas all other samples consist of a mixture of anatase, brookite and rutile (see XRD patterns in Appendix A). Accordingly, their apparent band gaps vary between 2.7 and 3.2 eV depending on their precise phase composition. Scanning electron microscopy (SEM) images of the TiO_2_ samples have been included as Appendix A). They show that, in general, the samples present an irregular morphology, but when a high HCl concentration is used in the synthesis, a spherical shape is developed (see Appendix A).

For this series of samples, a correlation between propene conversion and textural properties (surface area and total pore volume) was observed in a previous study [12]. The photocatalytic activity for propene oxidation was also found to depend on TiO_2_ crystallinity and phase composition, the anatase content being a relevant parameter. However, in the context of the present work, it should be emphasized that, since alkanes are less reactive than olefins, the rate-limiting step in alkane activation can be different [9]. This is in agreement with a study by Haeger et al., who demonstrated that the reaction mechanism of alkenes and alkanes is different [9].

### 3.2. Propane Oxidation in Photocatalytic Batch Reactor

#### 3.2.1. Propane Adsorption in Dark Conditions

Propane adsorption in the absence of illumination was evaluated after 10 min of exposure of propane to titania (named as “propane dark”) and the determined adsorption capacities are summarized in Figure 1 (see the data in Appendix A).

Two blank experiments (without catalyst) were carried out to ensure the reliability of the obtained data (Appendix A). The results are fully reproducible and, therefore, the difference between the initial propane concentration (4890 ppm) and “propane dark” for each sample is assumed to be the amount of propane adsorbed on that photocatalyst.

A correlation between adsorption capacity and specific surface area (as it increases, more adsorption sites for the inhibitors should be present) of the photocatalysts has not been revealed (see Figure 1). However, it can be concluded that the presence of rutile is detrimental for propane adsorption and, in fact, the highest propane adsorption capacity is determined for samples containing anatase and a certain amount of brookite (TiO_2_–0.5M and TiO_2_–12M). The fraction of brookite also shows some detrimental effect on the amount of adsorbed propane, but there is not a strong correlation between both parameters. In any case, the influence of rutile on propane adsorption is higher (estimated as three times that of brookite). The low affinity of rutile for adsorption of many organic compounds is in agreement with literature [14,20,21]. Adsorption of alkanes on brookite is feasible, as has been previously reported [14,22], and agrees with the statements above.

The surface chemistry of the photocatalysts, studied by thermogravimetry [12] (Appendix A), has been also considered to explain the obtained results, but a clear relationship between the surface OH groups content and affinity for propane adsorption has not been found. This means that although OH groups may play a role: (i) the differences in OH groups’ contents are not large enough to evidence such influence on the process and/or (ii) the influence of other parameters, discussed in detail in this study, prevail over surface chemistry, in the range of values studied.

#### 3.2.2. Propane Conversion under Illumination

GC analysis of the exhaust of the reactor reveals that the main carbonaceous reaction product of the photo-induced oxidation of propane is CO_2_. Some other products, like ethylene, ethane, acetaldehyde, ethanol, acetone, ether, etc., have been detected, but only in very low concentrations. In fact, a CO_2_ selectivity ≥96% is obtained in all cases. Therefore, propane conversion has been calculated assuming complete oxidation (Equation (3)), according to Equation (4): (see the photocatalytic process in Appendix A).
C_3_H_8_ + 5O_2_*→* 3CO_2_ + 4H_2_O(3)
(4)$$Propane\;conversion\;\left(\%\right)=\frac{{\left[{\mathrm{CO}}_{2}\right]}_{t}}{3}\times \frac{1}{{\left[C_{3}H_{8}\right]}_{0}}\times 100$$
where [CO_2_]*_t_* is the concentration of CO_2_ in the analyzed gas stream after a certain illumination time (*t*) (thus, $\frac{{\left[{\mathrm{CO}}_{2}\right]}_{t}}{3}$ is the concentration of propane converted to CO_2_ at time *t*), and ${\left[C_{3}H_{8}\right]}_{0}$ is the initial concentration of propane.

Figure 2a shows the propane conversion versus illumination time. Up to an irradiation time of 30 min, similar propane conversion has been determined for TiO_2_–1M, TiO_2_–3M, TiO_2_–7M and TiO_2_–12M. TiO_2_–0.5M and, in particular, TiO_2_–0.8M are clearly less active. Interestingly, for longer illumination periods of up to 60 min, the TiO_2_–12M sample seems to deactivate and the highest conversion, of approximately 35%, is obtained for TiO_2_–3M. Commercially available P25 still outcompetes all synthesized TiO_2_–XM materials.

As explained above, the interaction of the gas phase with the photocatalysts in the absence of light leads to propane adsorption. Under illumination, propane can be also adsorbed on the catalyst’s surface, and thus the amount of adsorbed propane at any time, *t*, can be calculated as:(5)$$Propane\;as\;adsorbed\;species_{t}={\left[C_{3}H_{8}\right]}_{0}-\frac{{\left[{\mathrm{CO}}_{2}\right]}_{t}}{3}-{\left[C_{3}H_{8}\right]}_{t}$$
where ${\left[C_{3}H_{8}\right]}_{0}$ is the propane concentration in the inlet stream (4890 ppmv); and $\frac{{\left[{\mathrm{CO}}_{2}\right]}_{t}}{3}$ and ${\left[C_{3}H_{8}\right]}_{t}$ are, respectively, propane converted to CO_2_ (ppmv) and propane remaining in the outlet stream (unreacted propane, in ppmv) after a certain illumination time (*t*).

The amount of propane adsorbed at each illumination time expressed in mmol/g is shown in Figure 2b. This figure shows that the amount of adsorbed species slightly increases with illumination time, in agreement with a dynamic adsorption/oxidation process. Furthermore, the calculated data suggest that activity is inversely related with adsorption capacity (Appendix A). For example, TiO_2_–3M possesses the lowest adsorption capacity while enabling the highest propane conversion. This is also revealed by the carbon mass balance (CMB) shown in Appendix A.

Interestingly, a relatively low quantity of propane that can be adsorbed on the surface is indicative of a relatively high photocatalytic rate, inferring that photocatalysts with a high rutile content would be the most active. Although many other factors, such as particle size and light absorption, determine photocatalytic rates, the high activity of P25 might be related to its relatively high content of rutile. In fact, Van der Meulen et al. [7] suggest that there is a positive synergistic effect between anatase and rutile, and conclude that samples with a larger average anatase crystal size are the most active. Sample TiO_2_–3M, with a high rutile and brookite content, is among the most active in the TiO_2_–XM series, but its activity is likely limited because of the low anatase content. Samples TiO_2_–1M and TiO_2_–7M show very similar catalytic behavior because they have very similar phase compositions and band gap energies. The low activity of sample TiO_2_–0.8M could be explained by the combination of both a high propane adsorption capacity and relatively low anatase content. Finally, TiO_2_–0.5M and TiO_2_–12M adsorb significant amounts of propane as they primarily contain the anatase phase, rendering their catalytic activity for propane oxidation low.

### 3.3. Study by In Situ DRIFTS

As an example of the recorded DRIFTS spectra, Appendix A show data obtained for sample TiO_2_–12M under air and propane atmosphere. Appendix A summarizes the main adsorption bands. These results allow to asses that the photocatalytic oxidation of propane involves: (i) the consumption of OH groups present on the TiO_2_ surface, and (ii) the creation of a series of surface species, observed mainly in the region 1200–1800 cm^−1^, indicative of CO bonds in acetone, formate, carbonate or bicarbonate groups (oxidation intermediates) [7,18].

A zoomed-in view of the 1200 to 1800 cm^−1^ region is shown in Appendix A, summarizing the IR spectra of all samples analyzed. The band assignment in this region is summarized in Table 2.

Although the main patterns are similar in all spectra obtained, there are some significant differences between them, likely due to the abundance of certain surface species in the presence of different TiO_2_ phases [7,14]. The main differences can be observed in the band centered at about 1700 cm^−1^, due to acetone (species (1) in Scheme 1). For P25, a single band is observed. Instead, for all TiO_2_–XM samples multiple bands appear, which can be related to different interaction sites [26] or to the presence of formic acid on the catalyst surface [27].

Figure 3 shows the DRIFTS spectra obtained after 60 min of illumination for TiO_2_–0.8M and TiO_2_–3M, being chosen as representative for their low (TiO_2_–0.8M), and high (TiO_2_–3M) activity in propane conversion. DRIFT spectrum obtained for P25 is shown as reference.

As already mentioned, the band due to acetone in the spectra of TiO_2_–0.8M and TiO_2_–3M seems to be composed of several peaks (at least two at 1712 and 1690 cm^−1^). The intensity of this band varies in the three samples as: P25 > TiO_2_–3M > TiO_2_–0.8M, following the observed catalytic activity. The intensity of the vibrational bands corresponding to formate (HCOO^−^), carbonate (CO_3_^2−^) and bicarbonate (HCO_3_^−^) species (bands in the 1300–1600 cm^−1^ region) inversely correlates with photocatalytic activity in the order P25 < TiO_2_–3M < TiO2–0.8M. For carbonate and bicarbonate species, no specific trend has been observed, likely related to their occurrence as reaction intermediate or product due to their formation by interaction of the generated CO_2_ with the catalyst surface [11].

In order to correlate photocatalytic activity data and obtained DRIFT spectra for the entire data set, the peak height of the band at 1690 cm^−1^, assigned to surface acetone species (the first intermediate in the transformation of propane, Scheme 1) has been used, despite the difficulty of an absolute quantification of the DRIFTS data.

As shown in Figure 4, the intensity of such a peak is highest for P25, whereas for all the other samples it falls in a close range, which resembles the propane conversion results of Figure 2a.

The same type of analysis undertaken with the peak heights of bicarbonate (bic) and formate (form) species reveals that, in general, the surface concentration of these species is lower when the concentration of surface adsorbed acetone (ac) species is higher. To analyze this in terms of relative values, the peak height ratios form/ac (height of peak at 1556 cm^−1^/ height of peak at 1690 cm^−1^) and bic/ac (height of peak at 1438 cm^−1^/ height of peak at 1690 cm^−1^) have been calculated and plotted for each sample, and are compared with the propane conversion values after illumination for 60 min (Figure 5).

As clearly revealed by the data summary provided in Figure 5, there is an almost inverse relationship between the form/ac and bic/ac ratios and the photocatalytic activity (only in the case of sample TiO_2_–12M propane conversion is lower than expected). This means that intermediate species, like formate and bicarbonate, when present in excess as compared to acetone, are indicative of deactivation/inhibition of the conversion of propane to CO_2_. Similar trends have been found for different illumination times (10 and 30 min). As mentioned above, bicarbonate could be considered either as an intermediate species or as species formed during reaction with the produced CO_2_ and, thus, it can be regarded as a by-product of the desired reaction. The concentration of bicarbonate species is probably in equilibrium with the gas phase or adsorbed CO_2_.

These results are in agreement with literature [7,11,14,28], which states that, on the one hand, the transformation of bicarbonate, carbonate and formate species to CO_2_ is the slowest step in propane oxidation and, on the other, the accumulation of these species on the TiO_2_ surface inhibits the adsorption sites for O_2_ (hindering the photoxidation process) [11].

The results obtained show that the determined form/ac and bic/ac ratios are higher for the TiO_2_–XM samples than for P25. This seems to indicate that formate and bicarbonate species are more stable on the TiO_2_–XM samples than on P25, what can be probably related with their lower crystal size. Such an accumulation of surface formate or bicarbonate species, that can be considered responsible of lower activity, is the highest in TiO_2_–0.8M, followed by TiO_2_–0.5M and TiO_2_–1M, and it is moderate in the case of samples TiO_2_–3M, TiO_2_–7M and TiO_2_–12M (see Figure 5). In fact, excluding sample TiO_2_–12M, there is a good inverse correlation between propane conversion and the bic/ac and, particularly, form/ac ratios (see Appendix A).

Thus, the combination of material synthesis allowing for control of phase composition while maintaining a similar crystallite size, propane adsorption and conversion, and in situ DRIFTS analysis enables us to derive the following guidelines: beneficial propane conversion is achieved for samples showing a low adsorption capacity of propane and a high intermediate concentration of acetone, as detected by DRIFTS. This is best achieved for samples containing large quantities of rutile phase alongside with anatase phase. Among the samples synthesized, TiO_2_–3M fulfills the requirements best. The positive role of low adsorption capacity of organic species [14,20,21] and the synergic effect of photocatalysts with mixed phase composition have also been previously reported [7] and are in agreement with the data obtained for P25 in this study. These data additionally emphasize that larger crystallite sizes are beneficial for propane conversion, being P25 the one that better fits these conditions. These two samples, TiO_2_–3M and P25, have in common the fact that they contain the largest amount of rutile. Moreover, TiO_2_–3M presents the highest brookite content from the TiO_2_–XM series, which is also beneficial to limit propane adsorption on the photocatalyst.

## 4. Conclusions

The photocatalytic oxidation of propane using anatase-brookite-rutile TiO_2_ samples synthesized by hydrothermal methods has been studied in a batch reactor and also by means of in situ DRIFTS analysis. Propane is adsorbed on the TiO_2_ samples in the absence of light. The amount of adsorbed propane is strongly and inversely related to the rutile content, and also to brookite content to a lower extent. It is highlighted that the photocatalytic activity is strongly influenced by the phase composition, and inversely correlated to propane adsorption. DRIFTS analysis revealed the existence of adsorbed acetone, formate, acetate and bicarbonate species. These intermediates (or by-products) are of importance for the rate of photo-oxidation of propane and, in particular, it has been shown that the presence of formate and bicarbonate negatively affect the oxidation of propane to CO_2_. The presence of predominantly small crystallites and/or low rutile (and maybe also brookite) content facilitates excessive formate and bicarbonate formation. The present study also shows that a combination of phases, especially anatase and rutile, introduces a synergistic effect. Finally, it is important to point out that the activities of the samples are not superior to that of P25 due to detrimental adsorption and formation of stable adsorbed species, which depend on the surface area and phase composition.

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
