# Peer review of "Photocatalytic Oxidation of Propane Using Hydrothermally Prepared Anatase-Brookite-Rutile TiO2 Samples. An In Situ DRIFTS Study"

_nanomaterials, 2020, doi:10.3390/nano10071314_

Round 1
Reviewer 1 Report
Casanova et al reported the photocatalytic oxidation of propane by using TiO2 NPs with complex phases and studied their oxidation process by in-situ DRIFTS. Although the results of the in-situ experiment are interesting, there is a lack of typical characterizations that should be addressed. After appropriate revision, I could consider this manuscript published in Nanomaterials. The specific comments are showed as followed:
- The motivation and statement of TiO2 with mixed phases used in this study are ambiguous, and their conversion performances are not superior to commercial P25.
- Under the dark conditions, the correlation between the adsorption activity and surface chemistry of the three phases should be explained clearly.
- The possible mechanism and charge transfer route of photocatalytic reaction by using TiO2 should be provided.
- What is the factor dominating the adsorption of intermediator then inhibiting the activity?
- The authors emphasized the phase composition of TiO2 NPs dominated the activity of propane oxidation. The related XRD pattern should be provided.
- For the completeness of the study, the morphology of various TiO2 by using SEM and TEM might be shown.
Reviewer 2 Report
The paper focuses on the removal of propane using hydrothermally synthesized TiO2. Moreover, intermediates byproducts are investigated in order to obtain a correlation between IR and propane oxidation.
In general, the results mostly support the authors' conclusions. However, some aspects of the manuscript must be carefully reviewed, discussed and improved.
1°) the originality, mechanism, and scientific reliability of the work are unclear. In my opinion, there are some major points that the authors should address:
2°) why do authors study the removal of propane (R–CO2 species formation ?) ?. the choice of concentration of this compound must be justified?
3°) the bibliographic part about carbonate-carboxylate-formate (R–CO2 ) species is too short, please give more information and other studies (line 52 page 2)
4° please add mores references about investigations on compound removal with photocatalysis. I suggest adding these refs (Chemical Engineering and Processing: Process Intensification 111, 1-6 ( 2017); Chemical Engineering Research and Design 106, 308-314 (2016))
5°) It is better to add flowsheet about the reactor. Please give the spectrum of irradiation
6°) In Figure 2. (a) about Propane conversion why all curves reached a maximum of around 30-40% (with the exception of TiO2-P25). The scientific explanation is needed
7°) the study of intermediates byproducts with photocatalytic and absorption proportions is very interesting but at any moment the authors discussed the results of the mass balance.
8°) what about the reusability of catalyst?
